# Anti-Cytomegalovirus Therapy: Whether and When to Initiate, Those Are the Questions

**DOI:** 10.3390/ph15070797

**Published:** 2022-06-27

**Authors:** Yumi Aoyama, Seiko Sugiyama, Takenobu Yamamoto

**Affiliations:** Department of Dermatology, Kawasaki Medical School, Kurashiki 701-0192, Japan; sugiyamaderma@med.kawasaki-m.ac.jp (S.S.); go-yama@med.kawasaki-m.ac.jp (T.Y.)

**Keywords:** cytomegalovirus reactivation, anti-CMV therapy, immune reconstitution inflammatory syndrome, dipeptidyl peptidase 4 inhibitor-associated bullous pemphigoid, autoimmune bullous disease

## Abstract

Cytomegalovirus (CMV) reactivation in patients with autoimmune bullous disease (AIBD) or severe drug eruption treated with immunosuppressive therapy was traditionally thought to be merely an epiphenomenon of the underlying immunosuppression. However, a detailed review of the clinical course of these patients revealed that CMV reactivation occurs upon rapid immune recovery, which is termed immune reconstitution inflammatory syndrome (IRIS), and that the timely initiation of anti-CMV therapy, when combined with maintenance doses of immunosuppressive agents, contributes to a rapid resolution of severe infectious complications thought to be refractory to conventional immunosuppressive therapies and unrelated to CMV reactivation. Thus, CMV reactivation resulting in fatal outcomes (CMV-IRIS) can be prevented by the timely detection of CMV DNA or antigens in the blood and by rapidly starting anti-CMV therapy while maintaining immunosuppressive therapy. Anti-CMV therapy is highly recommended for patients with CMV-IRIS or severe drug eruption who have risk factors for CMV reactivation resulting in fatal outcomes.

## 1. Introduction

Cytomegalovirus (CMV) reactivation during immunosuppressive therapy remains a major life-threatening complication in a substantial proportion of patients with an autoimmune disease or severe inflammatory disease, such as autoimmune bullous disease (AIBD) and severe adverse drug reactions [1,2,3]. Recent studies have argued that CMV reactivation is not uncommon even in previously immunocompetent individuals on immunosuppressive therapy and may be associated with high morbidity and mortality rates [4,5]. Thus, the identification of patients at risk for developing CMV reactivation is potentially of great clinical relevance. Patients with an autoimmune disease, such as AIBD, are generally considered to have an increased susceptibility to viral infection/reactivation, particularly herpesviruses [6]. In addition, herpetic superinfections can mimic erosive skin disorders that are indistinguishable from AIBD [6]. Herpetic infections should be suspected when patients with AIBD have an apparent flare or onset of other organ involvement unaccompanied by changes in serum autoantibody titers.

Although the impairment of immune responses was thought to be a prime risk factor for CMV reactivation, recent studies have revised this dogma and suggested that CMV reactivation may occur upon rapid immune recovery; inflammatory responses triggered by rapid immune reconstitution from an immunosuppressive state can lead to a series of immunologic reactions, which is termed immune reconstitution inflammatory syndrome (IRIS) [7,8,9]. Retrospective cohort studies have revealed that CMV reactivation can occur weeks or days before overt CMV disease, indicating that the CMV disease is developing long before the disease becomes clinically detectable [3,10]. Thus, recognizing CMV reactivation during immunosuppressive therapy as either a manifestation of IRIS or the result of immunosuppression would be key to the successful management of this complicated condition. 

Similar situations favoring CMV reactivation have also been reported in patients with severe drug eruption, which is termed drug-induced hypersensitivity syndrome (DiHS)/drug reaction with eosinophilia and systemic symptoms (DRESS) [2,11]. This syndrome represents a life-threatening multiorgan system reaction characterized by the sequential occurrence of herpes virus reactivation and is considered to be another manifestation of IRIS based on several similarities between both conditions [8,12]. Given that similar adverse events and heterogeneous outcomes are also observed in patients treated with immune checkpoint inhibitors and after COVID-19 [12,13], these adverse events might represent a common clinical phenotype, in which the exacerbation of inflammatory responses or paradoxical development of infectious disease occurs upon the reduction or cessation of iatrogenic immunosuppression, such as dipeptidyl peptidase 4 inhibitors (DPP-4i) [3,14,15], carbamazepine, allopurinol, pulse prednisone, cyclophosphamide, and prednisone, or during iatrogenic immunostimulation by immune checkpoint inhibitors. Although previous studies have indicated that anti-CMV therapy has a significant, positive impact on morbidity and mortality in DiHS/DRESS [2], debate continues about when to start and finish treatment.

In this article, we review four cases with AIBD admitted to our hospital between April 2011 and March 2017. Because CMV reactivation can be difficult to recognize in patients with AIBD or severe drug eruption and a delay in the diagnosis of CMV reactivation may increase the risk of serious morbidity and mortality, a disease flare unexplained by an increase in autoantibody titers should be considered as a manifestation of infectious IRIS. Therefore, we would like to demonstrate how effective anti-CMV therapy is in ameliorating severe complications, which were initially thought to be unrelated to CMV reactivation.

## 2. Results

### 2.1. Case Presentation

Case 1: A 71-year-old man with a history of type 2 diabetes and heart failure presented to our hospital with a 2-week history of multiple bullae in the oral cavity, on the lips, and around the eyes, as well as pruritic bullae over the trunk and extremities. He had been treated with linagliptin for 5 years. DPP-4i-associated mucous membrane pemphigoid (DPP-4i-MMP), Stevens–Johnson syndrome, and herpes simplex virus infection were suspected and linagliptin was immediately withdrawn. Despite the withdrawal of linagliptin, he again presented with worsening pruritic bullae and erosions. Physical examination revealed that 30% of his body surface area was covered with bullae, erosions, and crust, located on his trunk, genitalia, and bilateral upper and lower extremities with multiple erosions and bullae on the lips and oral mucosa (Figure 1). Biopsies obtained from inguinal lesions showed subepidermal bullae with sparse lymphocytic infiltration. Direct immunofluorescence analysis of a skin lesion showed the linear deposition of IgG and C3 along the basement membrane zone (BMZ). Results of an autoantibody panel found elevated BP180 NC16a domain IgG levels (17.6 U/mL; normal < 9 U/mL). Indirect immunofluorescence examination of normal skin revealed IgG antibodies at the BMZ, while indirect immunofluorescence analysis of 1 mol/L NaCl-split skin revealed IgG reactivity with the epidermal and dermal sides of the split using serum obtained at the initial presentation. Because herpes simplex virus DNA was not detected in his saliva at the initial presentation, a diagnosis of DPP-4i-MMP was made. Virological examinations on day 5 after hospital admission revealed CMV antigenemia, confirming the reactivation of CMV presenting as severe stomatitis. We immediately started intravenous 12 mg/kg ganciclovir (GCV) divided twice daily for 10 days until the antigenemia resolved. Systemic corticosteroid treatment (45 mg/day of prednisone) was initiated for MMP. A slight improvement of the skin lesions was detected, but the stomatitis deteriorated progressively over the course of 2 weeks (Figure 1), so intravenous 500 mg pulse prednisone for 3 days was administered on day 30 after hospitalization. Upper gastrointestinal endoscopy revealed multiple ulcers in the esophagus, which were suspected to be CMV reactivation and MMP, and a biopsy was performed. Subsequent immunohistochemical examination for CMV antigens revealed multiple cells with CMV inclusions and CMV antigen expression, thus confirming the histopathological diagnosis of CMV disease (at 35 days after hospitalization). Laboratory evaluation was remarkable for an elevation of white blood cells (15.0 × 10^9^ cells/L) with 80% neutrophils and 250.0 × 10^9^ cells/L platelets. The patient was immediately treated with GCV again. After we started combination therapy of 45 mg/day of oral prednisone and GCV, the bullae and erosions resolved completely within 7 days. With this combination therapy, his prednisone dosage was successfully tapered to 40 mg. Repeated esophagogastroduodenoscopy (EGD) and monitoring of CMV reactivation have revealed no reactivation. The patient was discharged 1 week after the bullae and erosions were resolved on a slow steroid taper. The patient’s skin disease remained unremarkable on a maintenance dose of 10 mg/day of prednisone for 3 months after discharge. He was then lost to follow-up. 

Case 2: The patient was an 85-year-old man who was diagnosed 3 years earlier with pemphigus vulgaris. The pemphigus was controlled with 5 mg prednisone for over 1 year; however, a few blisters flared up 1 month before he visited our hospital. He presented with herpes zoster at our department and was treated with 750 mg/day of acyclovir for 7 days and prednisone was increased to 15 mg/day for pemphigus. However, after herpes zoster, he developed widespread multiple blisters over the entire body surface with areas of sparing on the hands and lower extremities. Enzyme-linked immunosorbent assays showed the presence of IgG antibodies to desmoglein-3 (Dsg3) (1430 index value, iv) and Dsg1 (326.9 iv). Direct immunofluorescence analysis showed IgG deposition in the intercellular space in the epidermis. Intravenous immunoglobulin therapy (400 mg/kg daily) was administered for 5 days and followed by intravenous 500 mg pulse prednisone for 3 days. After pulse prednisone, intravenous immunoglobulin therapy was performed again. A significant decrease of serum anti-Dsg1 and -Dsg3 antibody levels was noted; however, the skin lesions were refractory to treatment (Figure 2). At 6 days after pulse prednisone, he developed pneumonia. One week before the onset of pneumonia, CMV antigenemia was detected. Coincident with or prior to the detection of CMV antigenemia, a decrease in platelet and white blood cell counts and an increase in aspartate transaminase and alkaline phosphatase levels were noted. Because a viral assay of sputum was not performed, anti-CMV therapy was not initiated despite compelling circumstantial evidence of CMV reactivation involving his pneumonia lesions and pemphigus vulgaris lesions. Virological examinations at 3 days after pulse prednisone revealed CMV antigenemia, again suggesting CMV-related pneumonia. Despite the decrease in anti-Dsg1 antibody titers, anti-Dsg3 antibody titers remained elevated. Due to this, prednisone was again increased to 50 mg/day without improvement. The patient thereafter suffered septic complications and eventually died.

Case 3: A 47-year-old man was referred to our department with widespread multiple blisters. His previous physician diagnosed him as having bullous pemphigoid (BP) and prescribed 60 mg/day of oral prednisone. However, BP was not controlled. His BP disease area index (BPDAI) was 50. A biopsy obtained from the trunk showed subepidermal bullae with eosinophilic infiltration. Direct immunofluorescence analysis of the lesions showed IgG deposition on the BMZ. Serum was positive for IgG and IgE autoantibodies against the BP180 NC16a domain. Two courses of intravenous 500 mg pulse prednisone for 3 days and 100 mg/day cyclophosphamide in combination with plasma exchange were performed. The skin lesions improved with a BPDAI of 19 with a reduction to 40 mg prednisone. After the withdrawal of cyclophosphamide, the patient experienced a severe relapse of BP with a rapid increase of the BPDAI to 67 (Figure 3). He again received two courses of pulse prednisone combined with 60 mg/day of prednisone followed by plasma exchange. His BP responded well to this regimen; at the end of the first month of treatment, the dosage of cyclophosphamide was reduced to 50 mg/day. At 10 days after the reduction of cyclophosphamide, CMV antigenemia was detected and anti-CMV therapy (GCV) was initiated. After the withdrawal of GCV therapy, he experienced acute onset of pneumonia. At this time, CMV antigenemia was detected again. The diagnosis of CMV pneumonia was made and GCV therapy was restarted. GCV therapy was continued until the patient was negative for GCV antigenemia. After the discontinuation of cyclophosphamide, the patient was placed on a gradually tapered dose of prednisone (Figure 3). The blisters healed gradually with no scarring. He died of Epstein–Barr virus-related primary central nervous system lymphoma.

Case 4:An 80-year-old woman presented with blistering skin lesions on her hands, feet, and mouth. Her primary care physician prescribed 30 mg/day of oral prednisone, which resulted in mild control of the blisters. However, 1 week after tapering prednisone to 15 mg/day, she experienced a marked worsening of blisters on her face, neck, back, bilateral upper and lower extremities, and oral mucosa (Figure 4). Examination revealed a 5 mm hemorrhagic bulla overlying edematous mucosa and an atrophic erosion with a ragged border on the buccal mucosa. No other ulcers were detected in the oropharynx. Enzyme-linked immunosorbent assays showed the presence of IgG antibodies to Dsg1 (342 iv) and Dsg3 (868 iv). Direct immunofluorescence analysis of the mucosa showed the deposition of IgG and C3 on the surface of keratinocytes. A diagnosis of pemphigus vulgaris was made and treatment with 50 mg oral prednisone was initiated. The bullous lesions on the trunk and extremities improved, but the oral lesions persisted. Additional bullae appeared on the right buccal mucosa despite a decrease in her serum anti-Dsg1 and -Dsg3 antibody levels. Based on the presence of persistent stomatitis, inability to taper prednisone over the course of 2 weeks, and detection of CMV antigenemia, CMV reactivation was suspected as the cause of the oral lesions when an apparent flare was not accompanied by changes in serum anti-Dsg IgG titers. The patient was treated intravenously with 12 mg/kg GCV divided twice daily for 1 week. Following this combination therapy of oral prednisone and GCV, the bullae and erosions on the oral mucosa improved gradually, although CMV stomatitis was noted during a prednisone taper. Without further GCV treatment, prednisone was tapered down to 15 mg/day. 

A brief description of Cases 1–4 has been presented previously [3].

### 2.2. Findings Suggestive of CMV Reactivation in AIBD Patients with Esophagitis and Pneumonia

An episode of CMV reactivation is defined by the presence of CMV DNA (≥20 genome copies in 10^6^ peripheral leukocytes) or CMV-C10/11 antigenemia in whole blood but not by the detection of increasing CMV IgG titers. Our cases indicate the importance of endoscopy in assessing esophago-gastroduodenal lesions in patients with AIBD on immunosuppressive therapy, even if they do not complain of odynophagia, dysphagia, or occasional gastrointestinal bleeding; esophageal involvement has been suggested to become evident even in the absence of the cutaneous manifestations of AIBD when the latter is in remission [16,17]. Endoscopic findings in most AIBD patients with esophageal involvement show multiple linear erosions and shallow ulcers with exfoliation of the esophageal mucosal tissue and friable mucosa that bleeds easily on touching. To distinguish AIBD patients with esophageal lesions complicated by CMV reactivation, clinicians should be familiar with the macroscopic appearance of CMV esophagitis, as shown in Case 1: typical lesions demonstrate multiple large ulcers covered with a layer of yellowish slough with erythematous and friable mucosa and some ulcers with raised whitish borders. In these lesions, histological and immunohistochemical examinations are needed to confirm CMV esophagitis. Histological analysis of esophageal lesions demonstrates large brick-red intranuclear inclusions and granular intracytoplasmic inclusions with CMV immunostaining. 

In contrast with CMV esophagitis, a definitive diagnosis of CMV pneumonia depends on the documentation of CMV infection in lung tissue; however, this is not usually performed because it is highly risky to perform a lung biopsy. Thus, the diagnosis of CMV pneumonia is often made based solely on the timely detection of CMV antigenemia at the onset of pneumonia without performing a lung biopsy and the detection of CMV DNA in sputum. Indeed, Case 3 was treated with 10 mg/kg GCV daily for 10 days once the diagnosis of CMV pneumonia was established, resulting in the complete resolution of pneumonia, consistent with the diagnosis of CMV pneumonia. In contrast, Case 2 was suspected of CMV pneumonia but was not treated immediately with GCV, resulting in a fatal outcome. These findings suggest that CMV pneumonia could represent a life-threatening condition in AIBD patients on immunosuppressive therapy when occurring as a manifestation of IRIS (CMV-IRIS).

In AIBD, especially pemphigus vulgaris, mucosal involvement is not limited to only the oral cavity but also extends to the esophagus or other gastrointestinal tissue, which is most commonly assessed using EGD. Indeed, esophageal involvement is reported to be observed in about 67% of patients with pemphigus vulgaris examined by EGD, but this is usually asymptomatic; thus, asymptomatic esophageal involvement is likely to be more prevalent than previously understood. Our findings emphasize the importance of recognizing esophageal and pulmonary involvement in AIBD patients with CMV reactivation as serious complications of AIBD. Unfortunately, only a handful of cases [18] with CMV esophagitis have been reported in patients receiving immunosuppressive therapy due to the lack of immunohistochemical examinations for CMV reactivation in the lesions. Our two patients with CMV esophagitis underwent EGD and endoscopy biopsies for the evaluation of CMV reactivation and were diagnosed with CMV esophagitis; importantly, both patients improved with anti-CMV agents in combination with maintenance doses of corticosteroids without additional aggressive immunosuppressive therapies. These findings illustrate the importance of achieving a balance in preventing rapid immune recovery that enhances CMV reactivation while reducing viral burden, as described later. If the esophagitis in Case 1 was viewed as a mere manifestation of AIBD and no tests were performed to identify CMV reactivation, the disease may have remained classified as AIBD-related symptoms. Thus, clinicians should have a low threshold for suspecting CMV reactivation in mucosal inflammation and ulceration in AIBD patients on immunosuppressive therapy and immediately consider antiviral therapy without reducing the doses of immunosuppressive agents, which appears to be intuitively rational.

### 2.3. Factors Contributing to the Onset of CMV-IRIS 

Although IRIS is typically viewed from the perspective of human immunodeficiency virus-positive patients treated with antiretroviral therapy, other situations associated with the rapid restoration of immune responses are also capable of triggering IRIS. Indeed, IRIS has been shown to occur in 14% of *Mycobacterium tuberculosis*-infected transplant patients upon the withdrawal of immunosuppressive therapy [19]. This situation is probably more clinically relevant in patients with AIBD on immunosuppressive therapy because AIBD patients are likely to be exposed repeatedly to the reduction or withdrawal of immunosuppressive therapy. Among the variety of available immunosuppressive therapies, pulse prednisone is the leading candidate for the rapid restoration of immune responses because a great reduction of prednisone dose over 1 week is necessary during tapering. Indeed, in Cases 1 and 2, CMV reactivation occurred or was exacerbated after pulse prednisone. Cyclophosphamide, when used as pulse therapy or at higher doses, may serve to provide one means by which IRIS may be triggered upon its withdrawal. Although DPP-4i are generally only mildly immunosuppressive, the dysregulated reconstitution of immune responses following DPP-4i withdrawal would be sufficient to induce CMV-IRIS, as shown in Case 1. During treatment with DPP-4i, in addition to general immunosuppression, the immunological environment would shift in favor of a Th2 immune response because CD26/DPP-4i is preferentially expressed on Th1 cells. This shift may provide a more favorable environment for CMV reactivation. A reversal in this balance to a Th1-dominated response by the withdrawal of immunosuppressive agents would be sufficient to induce CMV-IRIS.

### 2.4. CMV Reactivation as a Manifestation of IRIS and Anti-CMV Therapy

We showed that CMV reactivation occurred immediately after, but not during, the reduction or withdrawal of immunosuppressive agents in all cases examined, as Mizukawa et al. demonstrated in DiHS/DRESS [2]. In our AIBD cases complicated by CMV reactivation, immunosuppressive therapy was intensified with no improvement of symptoms before the detection of CMV reactivation. However, CMV reactivation was completely resolved by the immediate administration of anti-CMV agents while maintaining the doses of immunosuppressive agents. These results indicate that CMV reactivation occurring in AIBD patients on immunosuppressive therapy is a manifestation of IRIS. This emphasizes the importance of the prompt recognition of CMV reactivation in patients with AIBD.

Our four patients with AIBD (mean age ± SD, 70.8 ± 16.9 years) experienced CMV reactivation at a mean of 19.3 days (range, 6–35 days) after initial presentation; all patients were receiving prednisone and one was given cyclophosphamide. When rapid immune recovery is defined as a >50% increase in the frequency of lymphocytes within 1 week before the detection of CMV reactivation, CMV reactivation occurred coincidentally with immune recovery in all of our cases. Anti-CMV therapy was initiated after positive PCR or antigenemia results were obtained and continued until the detection of negative PCR or antigenemia results. The time to the initiation of anti-CMV therapy after the detection of CMV reactivation was 5 ± 4.7 days (range, 2–12 days). For Cases 1 and 4, anti-CMV therapy was started within 2 days after the detection of CMV reactivation, while for Cases 2 and 3, anti-CMV therapy was initiated at >3 days after the detection of CMV reactivation. In Case 3, CMV reactivation occurred again 2 days after cessation of anti-CMV therapy, indicating the need for longer anti-CMV therapy. Cases with AIBD who later developed CMV-related complications such as pneumonia and esophagitis tended to have CMV reactivation earlier in the course of AIBD after rapid immune recovery compared with those who did not have CMV-related complications. We have used ganciclovir as CMV therapy in four patients with good antiviral efficacy. However, we experienced a case in which ganciclovir could not be used long-term due to its myelosuppressive effect (Case 4). In this regard, the less toxic letermovir has an advantage.

## 3. Discussion

### 3.1. How to Predict CMV-IRIS

In our recent retrospective study on the outcomes of DPP-4i-BP after drug cessation, we demonstrated that the neutrophil-to-lymphocyte ratio (NLR) at baseline (before cessation) was significantly higher in an infectious IRIS group than in a group without complications [14]. It is also important to determine whether inflammatory responses occurring after DPP-4i cessation could result in changes in the proportion of blood cells and the NLR. The rationale for analyzing the change in the NLR after DPP-4i cessation is based on our findings that acquired hemophilia occurred as a manifestation of IRIS in a patient with DPP-4i-BP in association with the cessation of DPP-4i and prednisone [20] and that sudden exacerbations of DPP-4i-BP associated with CMV esophagitis (Case 1) occurred shortly after the cessation of pulse corticosteroids. Because the rapid recovery of lymphocytes observed in IRIS occurring after DPP-4i cessation could be reflected by a decrease of the NLR during the post-cessation stage, we could interpret a decrease in the NLR at short-term follow-up intervals as indicating that robust immune recovery as a manifestation of IRIS would occur late after the cessation of DPP-4is. This suggests the hypothesis that a decrease in the NLR after DPP-4i cessation reflects a late-onset IRIS status. Indeed, our analysis indicated that a decrease in the NLR during a short-term follow-up period was associated with a worse prognosis, while minimal changes in the NLR during the same period (the remission group) were associated with beneficial outcomes. This result indicates that although the NLR at baseline would represent a “snapshot” in which neutrophil dominance as a measure of innate immune competence is observed, changes in the NLR during a short-term follow-up period would represent a dynamic process, and the balance among various cell types likely changes over time. Thus, a decrease in the NLR may suggest that there is a tendency for the rapid recovery of lymphocytes after DPP-4i cessation, indicating that patients may develop a variety of manifestations of autoimmune IRIS ranging from acquired hemophilia to autoimmune thyroiditis.

Importantly, a decrease in the NLR during a short-term follow-up period was well associated with an increase in serum globulin levels and anti-BP180 NC16a antibody titers, which was typically observed in the autoimmune IRIS group. Consistent with this result, we recently demonstrated that there was a marked increase in serum globulin levels during the subacute phase (2–3 weeks after onset) after the cessation of the causative drugs in patients with DiHS/DRESS who later developed autoimmune disease [21], during which time regulatory T cells became functionally defective [22]. Thus, not only the NLR at baseline but also its changes during the post-cessation period would be useful for predicting the risk of developing autoimmune disease, especially when combined with changes in serum globulin levels during the post-cessation period.

Although our data demonstrate a positive correlation between the NLR and a worse long-term outcome, we do not know if our findings could be extended beyond DPP-4i-BP to include “usual” BP and IRIS in other immunosuppressed settings.

### 3.2. How to Treat CMV-IRIS with Anti-CMV Therapy and Immunosuppressive Therapy

CMV reactivation could aggravate the course of AIBD in patients who either fail to respond or experience AIBD exacerbations despite aggressive immunosuppressive therapy. Earlier studies recommended that immunosuppressive agents be tapered as quickly as is safely possible if immunosuppression was responsible for CMV reactivation. However, contrary to these earlier recommendations, we did not taper prednisone, but rather increased its dose (Case 1) when CMV reactivation was documented; our decision regarding corticosteroid therapy was the opposite of what would be expected if CMV reactivation was simply a response to immunosuppression. Because we recognized the occurrence of CMV reactivation in the course of AIBD on immunotherapy as a manifestation of IRIS, we neither withdrew nor reduced the doses of immunosuppressants when CMV reactivation was identified and immediately started anti-CMV therapy, resulting in the resolution of CMV-related complications, such as CMV esophagitis and pneumonia. Indeed, in all cases, CMV reactivation occurred within 1 week after the detection of a >50% increase in the frequency of lymphocytes. In support of this possibility, a fatal outcome occurred only in Case 2, for whom anti-CMV therapy was not initiated despite compelling circumstantial evidence of CMV reactivation. Thus, the data generated by this study indicate that when CMV reactivation is found in patients with AIBD on immunosuppressive therapy, anti-CMV therapy should be started immediately while maintaining the doses of immunosuppressive agents at least for 2 weeks because this regimen would have a long-term benefit for the patients, as shown in Cases 1, 3, and 4. We recommend that anti-CMV therapy not be withdrawn until having two negative confirmatory CMV tests, such as the presence of CMV DNA or antigenemia in the blood. According to recent studies on severe drug eruption associated with severe complications [2], most cases in whom anti-CMV therapy was started within 2 days after the detection of CMV reactivation recovered fully, indicating the clinical benefit of starting anti-CMV therapy immediately. In contrast, a delay in treatment, defined as an interval of ≥2 days from the first detection of CMV reactivation, was associated with the late development of CMV disease or complications not thought to be related to CMV reactivation; CMV disease or complications developed at 2 days after the cessation of anti-CMV therapy during the late phase of the disease in some cases. Considering the absence of CMV disease and complications during anti-CMV therapy and their transient resurgence soon after the cessation of anti-CMV therapy, earlier initiation and a longer duration of anti-CMV therapy after the detection of CMV reactivation are crucial for preventing the progression to CMV disease and serious complications. In ongoing validation studies with multiple centers, CMV reactivation was found to be a clinically relevant cause of severe fatal complications in the late stage of the disease, even if the complications appeared to be unrelated to CMV reactivation. Thus, we can conclude that fatal complications such as sepsis and pneumonia occurring in the late stage of the disease could be prevented by anti-CMV therapy, but only when it is started early and administered for a sufficient duration. 

The main limitations of our findings are the small number of patients examined from a single institution and the retrospective observational nature of our study. Thus, our results cannot be generalized easily. To confirm our findings, it is crucial to conduct larger prospective multicenter studies, with the aim of helping the design of appropriate therapeutic interventions.

### 3.3. Recent Topics on the CMV–Host Interactions Relevant to CMV Reactivation

In the last part of this article, we also briefly discuss recent topics on the CMV–host interactions, which give the reader a comprehensive overview of the current knowledge of CMV infection/reactivation in various situations. To expand research on the mechanism, the reader is invited to review these references [23,24,25,26]. Natural killer (NK) cells play a crucial role in host defense against CMV infection through the interaction of their surface receptors, such as the activating and inhibitory killer immunoglobulin-like receptors (KIRs) expressed on the surface of NK cells and T-cell subsets, and human leukocyte antigen (HLA) class I molecules expressed on their targets [23], suggesting that KIR and HLA polymorphisms play a primary role in protection against CMV. Consistent with this view, the most common clinical manifestation of NK cell dysfunction has been associated with some conditions, where recurrent and often severe herpesvirus reactivations occur, such as DiHS/DRESS [27]. Thus, specific combinations of KIRs with their cognate HLA ligands could be associated with CMV disease in various settings of autoimmune diseases and cancer [24]. This was confirmed by several reports, including the finding that patients carrying more than one activating KIR gene (KIR BA or BB genotypes) showed a rate of CMV infection/reactivation significantly lesser (20%) than that observed in patients with KIR AA genotype (36%) (with KIR2DS4) as the only activating KIR. In addition, the γ marker (GM) allotype could be involved in the immunological control of CMV infection/reactivation. In this regard, Di Bona et al. demonstrated that besides GM17/17 and GM 23 carriers, HLA-C2 and HLA-Bw4 are independently associated with the risk of CMV symptomatic infection in a multiple logistic regression analysis [25]. Because GM genes could also cause conformational changes in the antigen-binding site in the immunoglobulin variable regions associated with anti-CMVgB antibody specificity, GM allotypes could also influence antibody responsiveness to CMVgB; anti-CMV gB antibody levels were highest in GM 17/17 homozygotes, intermediate in GM 3/17 heterozygotes, and lowest in GM3/3 homozygotes [26]. These alleles have also been recently implicated in cancer risk, such as breast cancer neuroblastoma and glioma risk [26]. To confirm these observations, monitoring specific dysfunctional NK cell subsets in peripheral blood should be conducted in larger cohort studies.

## 4. Concluding Remarks 

Contrary to the traditional view, aggressive treatments including pulse corticosteroids were paradoxically associated with an increased risk of CMV reactivation upon their reduction or withdrawal. Given the neglect in many studies of evaluating CMV reactivation and the lack of immunohistochemical identification of CMV reactivation in the cutaneous and mucosal lesions of AIBD patients on immunosuppressive therapy, the prevalence of CMV reactivation in AIBD is likely to be underestimated. In patients with AIBD on immunosuppressive agents, a rapid reduction or discontinuation of aggressive therapy would be a prime candidate for the later development of CMV-IRIS, and a more thorough evaluation of CMV reactivation is absolutely needed in order to reduce the risk of developing CMV-related complications. In cases where IRIS is suspected upon subsequent severe flaring of the disease after pulsed prednisolone therapy, a more thorough clinical evaluation of CMV reactivation should be performed immediately, with prompt initiation of anti-CMV medications. Awareness of CMV-IRIS in patients with AIBD treated with pulsed prednisolone or other aggressive therapies will allow for earlier diagnosis, treatment, and relevant follow-up.

## Figures and Tables

**Figure 1 pharmaceuticals-15-00797-f001:**
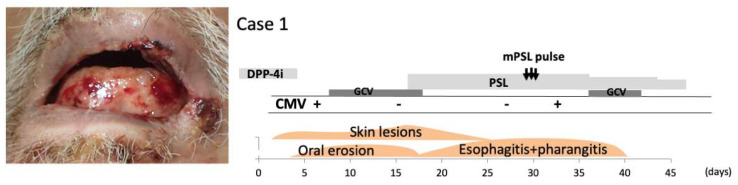
Case 1: Oral erosions caused by mucous membrane pemphigoid and its clinical course. Steroid pulse therapy triggered CMV antigenemia and esophageal erosions worsened.

**Figure 2 pharmaceuticals-15-00797-f002:**
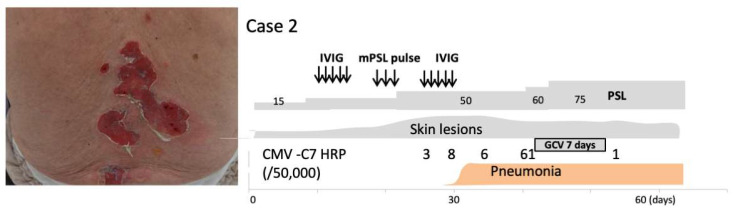
Case 2: Erosions due to pemphigus and its clinical course. CMV antigenemia developed following steroid pulse therapy, resulting in pneumonia.

**Figure 3 pharmaceuticals-15-00797-f003:**
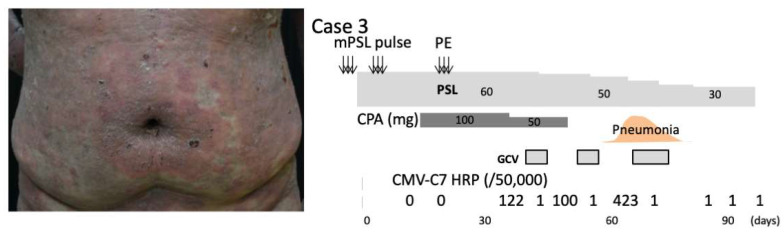
Case 3: Erythema caused by BP and its clinical course. CMV antigenemia developed following pulse therapy and plasma exchange, and the patient developed pneumonia.

**Figure 4 pharmaceuticals-15-00797-f004:**
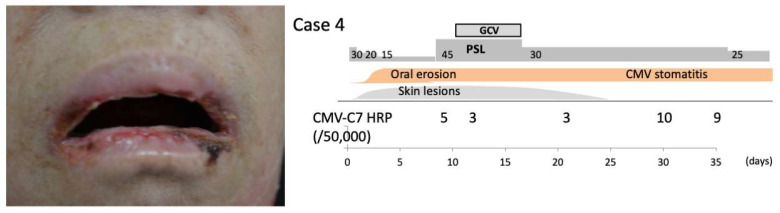
Case 4: Intraoral erosions due to pemphigus and its clinical course. Skin biopsy revealed CMV infection as a complication.

## Data Availability

Data is contained within the article.

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
