# Peer review of "Anti-Cytomegalovirus Therapy: Whether and When to Initiate, Those Are the Questions"

_pharmaceuticals, 2022, doi:10.3390/ph15070797_

Round 1

Reviewer 1 Report

I read with very interest the article entitled “Anti-cytomegalovirus Therapy: Whether and When to Initiate, Those Are the Questions”.

Author discusses about Cytomegalovirus (CMV) reactivation in patients with autoimmune bullous disease (AIBD) or severe drug eruption treated with immunosuppressive therapy by presenting four case report.

Overall the study is well-conceived and the manuscript well-written. The topic is really interesting.

However, some minor changes could improve the manuscript:

  • The manuscript presents several change font, mainly the number indicating references. Please carefully manuscript spell check to eliminate grammatical errors.
  • Page 4, line 11, please spelled out the full name of BP.
  • Page 5 subtitle “1 Findings suggestive of CMV reactivation….”, I think it is 2.2
  • In “2.5 How to predict CMV-IRIS” paragraph, Author discuss excessively about previous his studies, I suggest shortening this section. Moreover, I suggest to change “2.5 How to predict CMV-IRIS” and “2.5 How to treat CMV-IRIS…..” as Discussion section. In the text there are too many subparagraphs of results and no discussion
  • Page 9, Concluding remarks is written twice.

Finally, The content of the manuscript could be improve to better discuss about recent advances in the immunobiology of HCMV-host interactions highlighting the association between γ marker, human leucocyte antigens and killer immunoglobulin-like receptors and the natural course of human cytomegalovirus infection. A mention should be done to seminal works in this field: PMID: 24973460, PMID: 24737799, PMID: 29067686; PMID: 30764515.

Therefore, I believe that the present paper should be accepted after minor revision.

Reviewer 2 Report

The author presents a very nice and well written review on Cytomegalovirus (CMV) reactivation in patients going through an immune reconstitution inflammatory syndrome (IRIS). The work is well organized. English language and stlye are overall fine. I recommend to publish the review in the current form.

Author Response

Reviewer 2

The author presents a very nice and well written review on Cytomegalovirus (CMV) reactivation in patients going through an immune reconstitution inflammatory syndrome (IRIS). The work is well organized. English language and style are overall fine. I recommend to publish the review in the current form.

 Response

Thank you for taking the time to peer review and we appreciate the evaluation.

Reviewer 3 Report

This is a useful summary of four case reports for a relatively rare condition. However, the implications regarding the complex biology of CMV interaction with the host immune system may apply to higher incidence conditions including Long Covid and shingles (Herpes zoster).

The only details missing in the manuscript are the specific antiviral drugs used to suppress CMV. It would be of particular interest to know if letermovir is superior to ganciclovir, as the former results in less neutropenia. 

Author Response

Reviewer 3

This is a useful summary of four case reports for a relatively rare condition. However, the implications regarding the complex biology of CMV interaction with the host immune system may apply to higher incidence conditions including Long Covid and shingles (Herpes zoster).

Comment 1.

The only details missing in the manuscript are the specific antiviral drugs used to suppress CMV. It would be of particular interest to know if letermovir is superior to ganciclovir, as the former results in less neutropenia. 

Response

Thank you for your suggestion. Accordingly, we added the sentences as follows.

Revised text, p.7 line 300-303.

We have used ganciclovir as CMV therapy in four patients with good antiviral efficacy. However, we experienced a case in which ganciclovir could not be used long-term due to its myelosuppressive effect (case 4). In this regard, less toxic letermovir has an advantage.

Reviewer 4 Report

This case-series showed 4 cases of CMV reactivation in AIBD and performed a rapid review of reactivation of CMV.

Considering the type paper is well written, introduction and result are in line, the text is easly to understand.

Minor request:

Line 62, citation [2] has different format, please ceck;

line 149 please explain what BP stand for;

Line 237 citation [18] has different format, please ceck.

Author Response

Reviewer 4

This case-series showed 4 cases of CMV reactivation in AIBD and performed a rapid review of reactivation of CMV.

Considering the type paper is well written, introduction and result are in line, the text is easly to understand.

Minor request:

Comment 1.

Line 62, citation [2] has different format, please check;

Response

We corrected the font following your suggestion.

Comment 2.

line 149 please explain what BP stand for;

Response

We spell out of BP as follows.

Revised text p.4, line 148-149.

A 47-year-old man was referred to our department with widespread multiple blisters. His previous physician diagnosed him as having bullous pemphigoid (BP)

Comment 3.

Line 237 citation [18] has a different format, please check.

Response

We corrected the font following your suggestion.